# Empirically validating a computational model of automatic behavior shaping

**Vincent Berardi**[1]*, **Tian Lan**[2], **Ariane Guirguis**[2], **Uri Maoz**[1,2,3]

**1** Department of Psychology, Crean College of Health and Behavioral Sciences, Chapman University, Orange, California, United States of America, **2** Institute for Interdisciplinary Brain and Behavioral Sciences, Crean College of Health and Behavioral Sciences, Schmid College of Science and Technology, Chapman University, Orange, California, United States of America, **3** Division of Biology and Biological Engineering, California Institute of Technology, Pasadena, California, United States of America

* berardi@chapman.edu

**Data Availability Statement:** The software used to run the experiment, de-identified data, and scripts for statistical analyses are publicly available on Github (https://github.com/vancebee/autoshaping).

## Abstract

### Background

Mobile sensing technology allows automated behavior shaping routines to be incorporated into health behavior interventions and other settings. In previous work, a computational model was built to investigate how to best arrange automatic behavior shaping procedures, but the degree to which this model reflects actual human behavior is not known.

### Purpose

To translate a previously developed computational model of automatic behavior shaping into an experimental setting.

### Methods

Participants ($n = 54$) operated a computer mouse and attempted to locate a hidden, randomly-placed target circle on a blank computer screen and clicks within some threshold distance of the target circle were reinforced by a pleasant auditory tone. As the trial progressed, the threshold distance narrowed according to a shaping function until eventually only clicks within the target circle were reinforced. Accumulated Area Under Trajectory Curves and Time Until 10 Consecutive Target Clicks were used to quantify the probability of the target behavior. Linear mixed effects models were used to assess differential outcomes for concave up, concave down, and linear shaping functions.

### Results

In congruence with the computational model, concave-up functions most effectively shaped participants' behavior, with linear and then concave-down shaping functions producing the next best outcomes.

### Conclusion

Concave-up shaping routines most effectively generated target behavior, which should be confirmed in health behavior trials. The automatic shaping routines that this study helps

**Funding:** The author(s) received no specific funding for this work.

**Competing interests:** The authors have declared that no competing interests exist.

develop can be applied in a number of domains, including exercise intensity and duration, tobacco/cannabis smoking, caloric intake, and screen time.

## 1. Introduction

Operant conditioning posits that behaviors met with reinforcing consequences will be more likely to be repeated under similar contexts in the future, while behaviors met with punitive consequences will be less likely to be repeated [1]. These consequences can either exist naturally (e.g., regular exercise leads to better health), or they can be engineered by an external entity (e.g., a counselor provides a monetary incentive every time their patient meets the CDC guideline of 150 minutes of moderate intensity physical activity in a week). Regarding the latter, it can often be effective to gradually guide individuals toward a goal, especially if the objective is complex or difficult to achieve. In other words, instead of waiting for a target behavior to be emitted to provide a reward, as per simple operant conditioning, it might be useful to initially establish a looser goal that eventually converges toward the final goal. Operant theory specifies this process as *behavior shaping*, which is the cultivation of a targeted behavior via reinforcing successive approximations to the target [2]. During the shaping process, the threshold for reinforcement narrows with time, as behaviors are required to become ever more similar to the target behavior in order to qualify for reinforcement (i.e., for a reward; Fig 1a). This process often leads to complex behaviors that would otherwise not be emitted as quickly, or at all [3]. When implementing a shaping routine, practitioners must i.) discriminate the deviation between an emitted behavior and the target behavior and ii.) optimally determine when to refine the reinforcement criteria so that only closer approximations to the target are rewarded. Proficiency in these efforts arbitrates the ultimate effectiveness of a shaping procedure, with specialists (e.g., counselors and coaches) typically mastering these skills as an art [4].

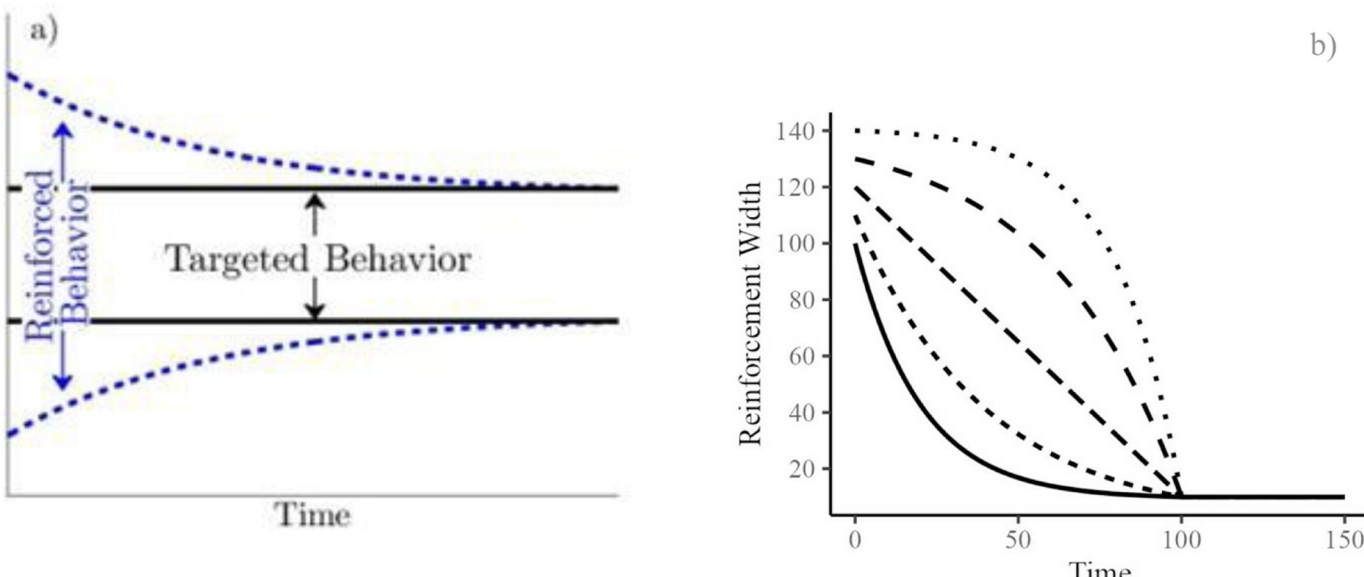

**Fig 1.** a) Schematic of a shaping procedure showing the window of reinforced behavior narrowing towards a target over time. b) Five examples of continuous shaping functions used in the computational model, governing the reinforced class size as it narrows with time towards a target size of 11. Note each function's different initial width (i.e., y-intercepts) and decay rate.

In general, health interventions have historically focused on antecedents of behavior (e.g., knowledge, prompts) rather than post-cedent features that enable operant phenomena like behavior shaping [5]. While behavior shaping has largely been implemented via applied behavior analysis to address development disorders [6], there have been calls for its integration it into more general settings [7, 8]. Shaping procedures that scale are of particular interest in the context of mobile health (mHealth) behavioral studies, which use smartphones, wearables, and other remote sensing devices to continually monitor health behaviors [9], allowing the similarity with a target behavior to be frequently assessed over extended periods of time. mHealth studies have also begun to implement contingency management systems that automatically reinforce participants when a target behavior is detected [10, 11]. The pairing of continual monitoring with programmed reinforcement has the potential to support an autonomous shaping paradigm that automatically detects behavior and reinforces participants according to criteria that becomes stricter over time. Such procedures can scale shaping routines within health interventions without relying on resource-intensive specialist coaches/counselors. For instance, smokers might have a target behavior of non-smoking and be automatically rewarded for abstinent time periods [11], with the time required for reinforcement steadily increasing. Similarly, individuals could be automatically reinforced for meeting daily physical activity goals that become more challenging over time.

Because automatic shaping is a nascent field, its potential alignment with mHealth is not widely recognized and the specifics of how to best arrange shaping procedures are unknown. For example, in the above hypothetical smoking example, we do not know how quickly the abstinent time periods that qualify for reinforcement should increase. Should one minute per day or thirty minutes per day be added? A small increase at the outset of the intervention followed by faster increase later, or vice versa? To demonstrate the potential effectiveness of automatic behavior shaping and to preliminarily investigate how to best arrange autonomous shaping procedures, Berardi et al. performed computer simulations of behavior shaping by extending a computational model of behavior dynamics to include a shaping component [12]. The original model [13] has been shown to capture many behavioral phenomena [14–18] and has selectionist reinforcement features that are uniquely congruent with behavior shaping. Within the Berardi et al. computational shaping experiments, a digital organism emitted a single behavior at each time step, where each potential behavior was represented by an integer between 0 and 999. The range of integers from 495–595 were defined as a *targeted class* that were reinforced when emitted (see Ref. [13] for details regarding the operationalization of reinforcement). To implement behavior shaping, at the outset of each simulation, a wider range of integers (i.e., behaviors) than the targeted class were reinforced, which was denoted as the *reinforced class*. Within each simulation, the width of the reinforced class contracted with time towards the targeted class, as governed by either a discrete or continuous shaping function.

Continuous shaping functions (examples of which appear in Fig 1b) were parameterized by two values: i.) the initial width of the reinforced class, given by the *y*-intercept of the function and ii.) the rate of decay towards the targeted class. Five hundred seventy-four unique parameter combinations were explored, which produced shaping functions that could be classified as concave down: slow initial descent followed by rapid descent to the target width; concave up: rapid initial descent followed by slow descent to the target width; or linear: constant descent to the target width. (See Fig 3 for exemplars of these three categories). These shaping functions differentially balance two competing effects: i.) narrowing the reinforcement window quickly enough as to avoid unduly rewarding and propagating non-targeted behavior and ii.) narrowing the reinforcement window slowly enough so that sufficient reinforcement to shape behavior is provided. For each shaping function parameter combination, 5,000 simulations were

averaged together to produce an empirical probability that a target behavior was emitted at each time step. Results indicated that continuous, concave up shaping functions were most successful at producing the targeted behavior, particularly early in the trial [12].

The computational results are compelling, but it is not straightforward to draw parallels between real-world scenarios and the computational model's focus on abstract behaviors coded as integers. Furthermore, the computational model relies on a similarity metric for behaviors that is defined by a difficult-to-translate mathematical operation (i.e. differences between integers). Therefore, the degree to which the predictions made by the computational shaping model generalize to human behavior is unknown. Establishing this link is a critically important step towards behavior shaping becoming a widely deployed technique in mHealth promotion interventions. To address this knowledge gap, we developed and performed a laboratory behavioral experiment that mimicked key features of the computational behavior shaping model. Our aim in performing this work is two-fold. First, we seek to highlight the general principles underlying behavior shaping and to outline considerations for its implementation (e.g. the continued updating of reinforcement criteria). Second, by establishing a correspondence between laboratory and digital experiments, we will demonstrate that the latter approximates real-world behavior shaping outcomes to a sufficient degree to make it a suitable tool for efficiently sandboxing behavior-shaping procedures before deploying them within health promotion trials.

## 2. Materials and methods

### 2.1 Ethics statement

This work was approved by the Chapman University Institutional Review Board (IRB-18-176 and IRB-20-59). Written informed consent was obtained from each participant.

### 2.2 Experimental design

Participants' task in the validation experiment was to locate and click within an invisible *target circle* hidden on a blank computer screen (see Fig 2). This target circle corresponded to the computational model's targeted class of integers. Analogous to the reinforced class in the computational model, a second concentric circle with a larger radius, the *reinforcement circle*, defined the initial reinforcement area. Participants were told to find the target area and when they clicked in the reinforcement circle, a pleasant tone was played; an unpleasant tone was played when participants clicked outside of that area. These auditory stimuli are consistent with those used in our previous mHealth behavior trials [19]. Automatic shaping was incorporated by contracting the reinforcement circle towards the boundaries of the target circle according to one of three shaping functions which were qualitatively similar to those used in the computational model. In other words, while our interest was in generating participant clicks within the target area, the experimental procedures began by reinforcing an area wider than the target, with this criterion becoming stricter with time until eventually only clicks within the target area were reinforced.

As was the case with the computational model, the contraction of the reinforcement circle towards the target circle was governed by the shaping function

$$r(n) = \frac{-300}{1 - e^{10b}} \left(1 - e^{bn}\right) + 400; \ 0 \le n \le 10 \tag{1}$$

which specified the radius $r$ of the reinforcement circle after the $n^{\text{th}}$ reinforcement with contraction rate $b$. This function was designed so that after 10 contractions, the reinforcement circle would contract to the size of the target circle [$r(10) = 100$]. The radius of the reinforcement

## Procedure for a Single Trial

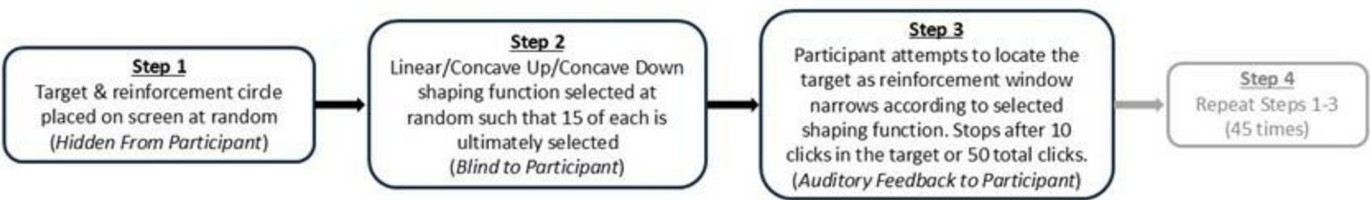

## Example Procedure for a Single Participant

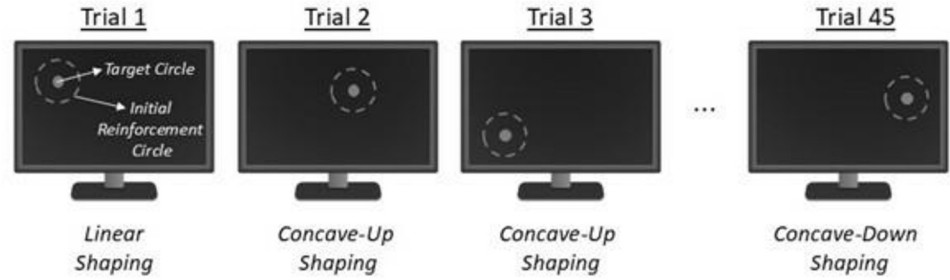

**Fig 2. Experimental procedure.** The target and reinforcement circles were randomly placed on the screen in each trial and were, critically, not visible to the participant. The participant clicked on the screen aiming to locate the target, with a pleasant tone provided if the screen location clicked was within the reinforcement circle and an unpleasant tone provided otherwise. As each trial progressed, the reinforcement circle collapsed towards the target circle according to either a linear, concave-up or, concave-down shaping function (15 of each type, with the order of the overall 45 trials randomly shuffled for each participant). These schematic figures show only the target/reinforcement circles and do not illustrate the shaping functions, which are shown in Fig 3.

circle was then held constant for the remainder of the trial. While the computational simulations tested 41 different values of $b$, replicating this in a real-world experiment would have been too resource intensive. Instead, we focused on three specific cases: i.) b = -0.3, representing a concave-up function; ii.) $b$ = 0.3, representing a concave-down function; and iii.) $b \approx 0$, which, after a linear Taylor expansion, corresponds to the function $r(n) = -30n + 400$. These cases capture three distinct patterns explored in the simulations, namely: i.) rapid initial narrowing (concave up), which leads to the target being encountered and reinforced as quickly as possible, but risks participants struggling to perform the target behavior and receive the associated reward; ii.) slower initial narrowing (concave down), which provides participants with a steady history of reinforcement that allows them to identify the antecedents of this outcome; and iii.) steady linear narrowing of the reinforcement window, which is a compromise between the first two phenomena. This approach allowed us to broadly compare computational and experimental results, while leaving more detailed investigations (e.g., slight versus extreme concave-down functions) for future research. See Fig 3 for an illustration of the shaping functions.

The trial was a within-subjects design (all participants experience each condition) where each participant completed 45 target location trials, which were randomly assigned to each of the three shaping functions, such that participants were exposed to each function 15 times. (Pilot data indicated that 15 exposures to each shaping function was sufficient to detect function-related differences in outcomes.) Therefore, the order in which the shaping functions were encountered varied across participants, who were not informed about the shaping procedure. At the beginning of each trial, the shared center of the target and reinforcement circles

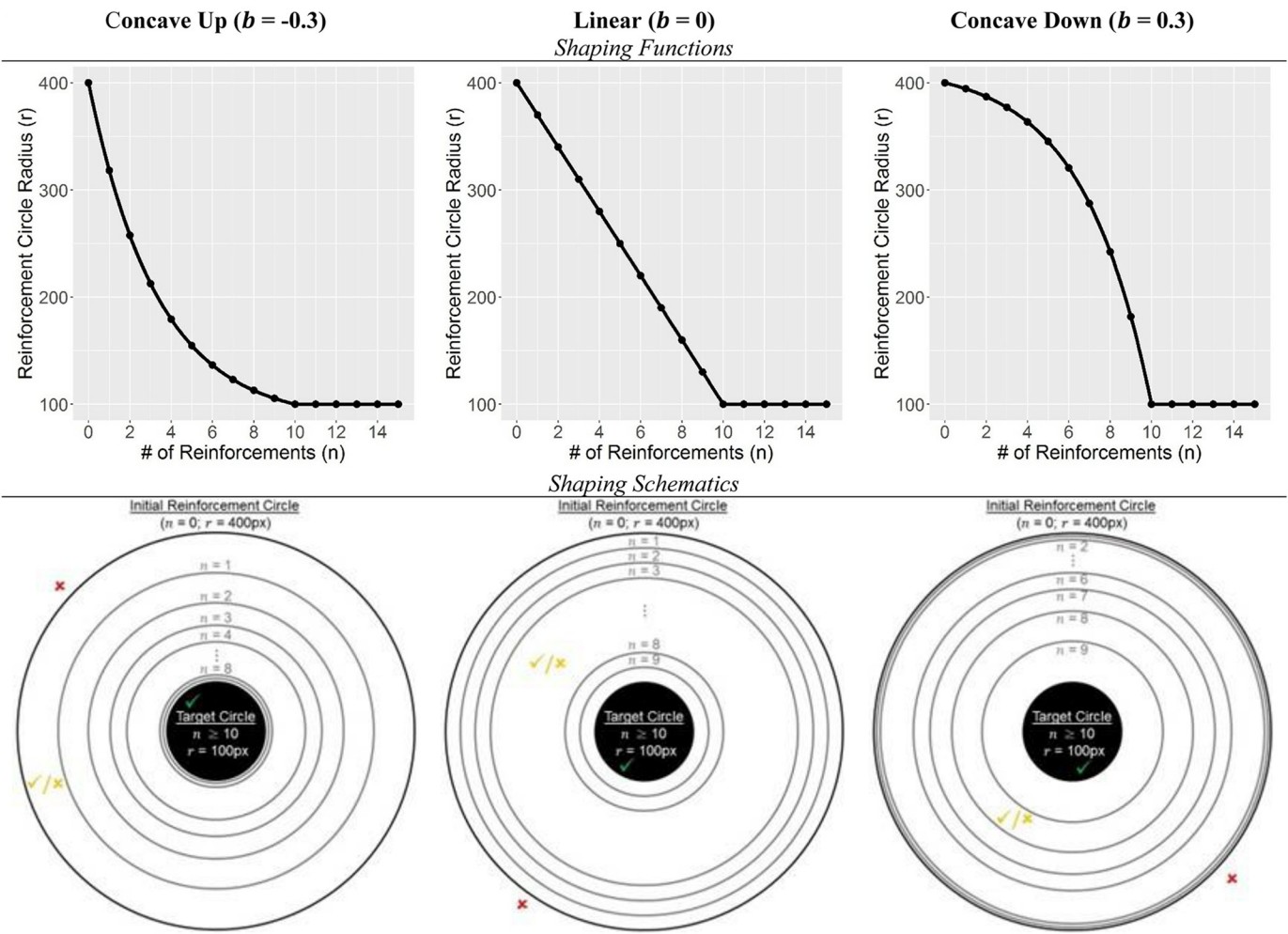

**Fig 3. Shaping functions with functional form and schematic shown in the first and second rows, respectively.** The shaping functions correspond to three different values of the contraction rate, *b*, in Eq (1): b = -0.3 (left), b ≈ 0 (center), and b = 0.3 (right) and illustrate the radius of the reinforcement circle as a function of the number of reinforcements. In the schematics, the black-filled circle is the target circle, and the larger circles are reinforcement circles, which start with a radius of 400px and contract toward the target circle after each reinforcement according to the shaping function. The green check box indicates a click that will always be reinforced (pleasant sound), since it is within the target. A click at the location of the red X will never be reinforced since it is never inside a reinforcement circle. A click on the location of the orange check/X may or may not be reinforced, depending on the size of the reinforcement circle at the time of the click. Each trial ended after 10 consecutive clicks in the target circle or 50 total clicks, whichever occurred first.

was randomly determined using Matlab's random number generator, with the constraint that the entire reinforcement circle remained within the screen. Pilot testing indicated that a 100 pixels (px) radius for the target circle and 400px radius for the reinforcement circle were appropriate. In contrast with the computational experiments, the starting width of the reinforced area was not systematically varied (i.e. it was fixed at 400px in every trial), since resources were limited and computational outcomes were not sensitive to this parameter. Each trial consisted of 50 clicks, but to prevent redundant clicking after the target was located, trials ended early if 10 consecutive clicks landed within the target circle.

### 2.3 Sample description and experimental procedures

Fifty-four undergraduate college students attending [*university name withheld to maintain confidentiality*] participated in the study. This sample size was based on data gathered from an

independent pilot study ($n$ = 12) where we tested study protocols and procedures. An a priori power analysis indicated that $n$ = 48 participants were sufficient to power our study to 0.9 with a significance level of 0.05. This was performed by considering participants' average number of clicks for each shaping function, fitting a within-factors, repeated measures ANOVA model to calculate the effect size (Cohen's $f$ = 0.33), and using G*Power for a power analysis associated with this model and effect size. We opted for $n$ = 54 participants, which allowed for a loss of data of up to 10% of participants while still adhering to our power analysis requirements. Note that data from pilot participants was not included in any of the analyses presented in this paper.

Participants were recruited between May 3, 2019, and November 11, 2019, using the SONA Experiment Management System (SONA), a university platform designed to recruit students from courses with research participation requirements as part of their curriculum. Eligibility criteria included enrollment in the university's SONA subject pool, the ability to understand English software instructions, and the absence of any visual or hearing impairments. Upon completing the study, participants received credit in line with their course policies. Of the 54 students in the study, 30 were female and 24 were male.

The experiment was conducted via a custom application built in MATLAB's Psychophysics Toolbox v3 deployed on a Dell Precision Workstation with a 24" Dell monitor, with a 1920 x 1080 resolution, an eye-to-screen distance of approximately 60 cm, and a refresh rate of 60 Hz. Each participant's trial was conducted individually and administered by a trained research assistant. After being provided with a brief description of the study, participants were consented and then sat in front of the study computer, which displayed a blank screen. Participants were shown a demonstration of how to operate the system and were then instructed to locate the randomly generated, hidden target circle during each trial. Participants were prompted to take a 3–5-minute break after completing the 15th trial, and again after the 30th trial. After some early participants were observed bypassing the breaks, the software was modified to make these breaks mandatory. Upon completion of the experiment, participants were debriefed and granted the extra credit for their courses via SONA.

### 2.4 Outcome metrics

**2.4.1 Area under the target trajectory.**   For the computational model that we are aiming to validate, results were quantified by considering the 5,000 simulations for each uniquely parameterized shaping function and calculating the empirical probability of a target behavior being emitted at each time point. This allowed a *target behavior trajectory* to be formed, with the area under the curve, or accumulation area (AA), for this trajectory serving as the main outcome measure. This process was replicated in the experiment by considering the time-ordered series of mouse clicks (i.e., Click 1, Click 2, . . ., Click 50) for each participant (See Fig 4a–4c), and calculating the empirical probability of a click being in the target circle, averaged over the 15 trials for each of the three shaping functions. For trials that ended early due to 10 consecutive target clicks before the 50th click, we assumed that the remaining clicks (up to 50) would have been target clicks and thus assigned them to the target (and designated them as red in the figure). The averaged target-click trajectory was smoothed via loess (bandwidth = 0.4) and then the AA was calculated (see Fig 4d). Higher AA values are associated with more effective shaping routines where the target behavior is attained more quickly.

**2.4.2 Total clicks.**   We also sought to assess the outcomes without requiring all trials for a given shaping category to be averaged. While aggregated trials allow the probability of a target click to be empirically calculated, single trials have a binary outcome (target vs. non-target) for each click, which does not support the calculation of a target behavior trajectory and associated

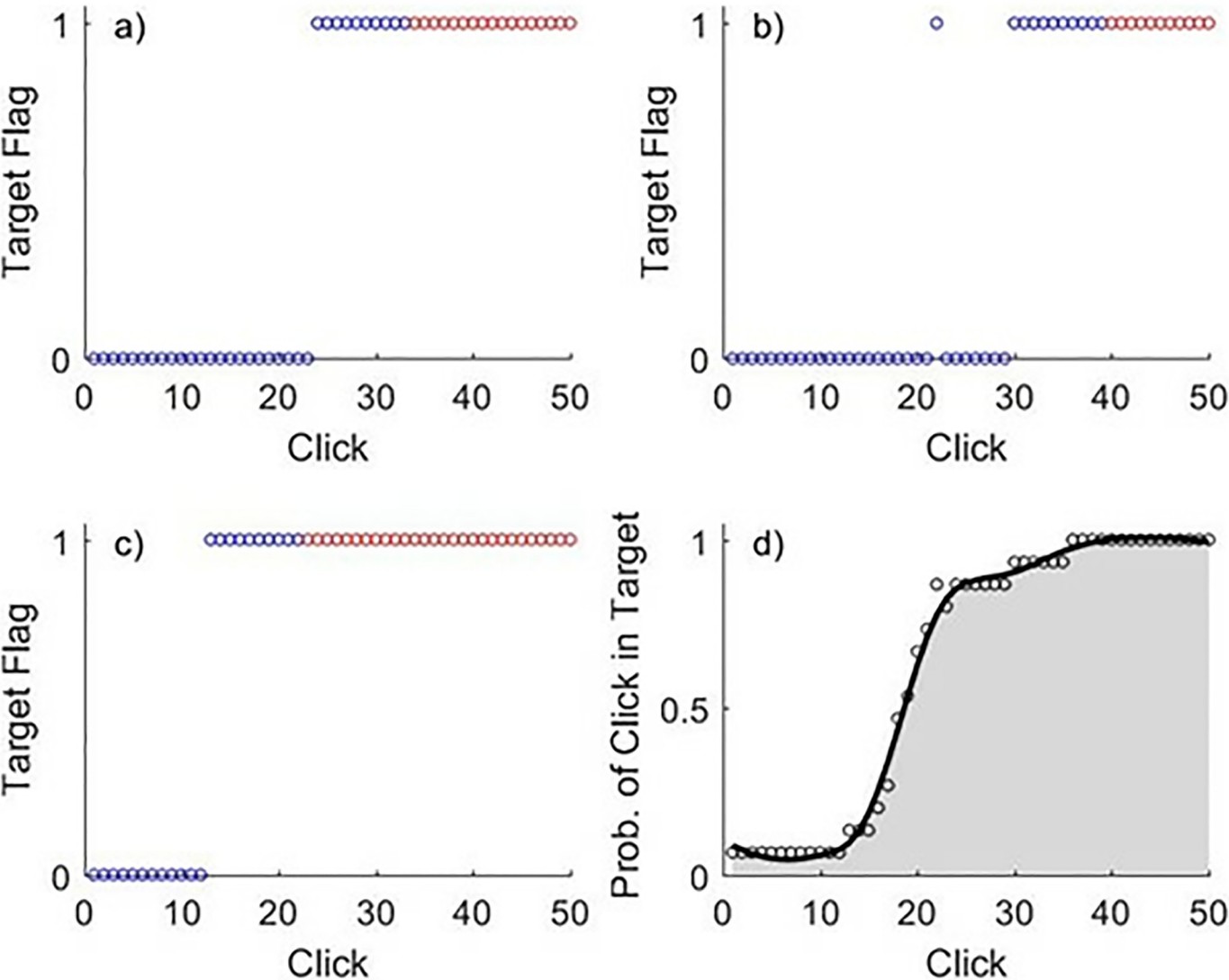

**Fig 4. AA calculation for the trial.** Panels a)-c) illustrate the click result for trials 5, 14, and 15 from participant 190503_04451, which are three of the fifteen concave-down trials for this participant. These trials were selected at random and are representative of process performed for all participants, trials, and shaping functions. The target flag for each click was set to 0/1 if the click was outside/inside the target circle. After 10 consecutive clicks within the target, we ascribed the remaining clicks until 50 to the target. The blue points represent actual clicks made by the participant, while the red points represent clicks that would have been carried out had the trial not stopped because the participant clicked 10 times consecutively within the target circle. Panel d) depicts the mean result for each click, averaged over all 15 concave-down trials. A smoothed line was fit to this average and accumulation area (AA) was calculated. This process was performed separately for linear, concave-down, and concave-up shaping functions for each participant.

AA, so instead we used the total number of clicks in each trial as an outcome measure. As shown in Fig 4a–4c., a smaller number of total clicks (in blue; not including clicks in red) corresponds to more effective shaping and a trajectory that jumps to 1 more quickly, therefore resulting in a higher AA. In other words, the AA and number of clicks per trial outcomes are inversely related, though not perfectly correlated.

### 2.5 Statistical analysis

To investigate differences in the outcome measures according to the shaping function, we implemented mixed effects linear modeling when AA was the independent variable. This

hierarchical regression approach is similar to repeated measures ANOVA, but more flexible in terms of missing data, clustered designs, and unbalanced measures. While these issues were not present in the current study, the more robust analytic approach was used in case such issues are encountered in the future. More specifically, using a random-intercept, Gaussian mixed-effects model, the shaping function was regressed on AA, with participant ID serving as the random effect. The variance partition coefficient was calculated, which indicates the proportion of total variance in the dependent variable that is accounted for by the shaping functions, which serves as a measure of effect size.

When regressing the total number of clicks on the shaping function, we initially fit a generalized mixed effects model as above (but with a negative binomial outcome due to the count nature of the dependent variable), with participant ID and trial number serving as completely crossed random effects. However, the proportion of variance explained by the random effects was < 0.01, so we elected to proceed with a more parsimonious generalized linear regression model without a hierarchical structure. While count variables can be modeled via Poisson regression, this approach assumes that the mean and standard deviation of the dependent variable are equal. The dispersion parameter can be used to assess this assumption, which for our model was 8.33. This value is much larger than the dispersion parameter near 1 that is expected for Poisson models and indicates overdispersion (variance larger than mean), making the negative binomial model appropriate. The effect size for this model is given by the IRR (incidence rate ratio), which represents how many more (IRR>1) or less (IRR < 1) clicks there were for one group versus another.

For both models, the normality of residuals was confirmed visually via Q-Q plots and density estimates of the residuals. The linear shaping function was set as the referent level in our models, and a contrast matrix was used to examine the difference between results for concave-up and concave-down shaping.

## 3. Results

Table 1 illustrates descriptive statistics for the two outcome variables, stratified by shaping function. For both outcome measures, the values obtained using the linear shaping function were similar to those averaged over all three shaping functions, while the concave down/up shaping functions produced outcomes roughly equidistant from the values seen for the linear function. The inverse relationship between these two outcomes can be seen by concave down (up) outcome producing a smaller (larger) AA and larger (smaller) number of clicks to reach the target.

Table 2 illustrates the fixed effects from the mixed effects regression models for both the AA and Total Clicks outcomes. This quantifies the statistical significance of differences in mean outcome values across the different shaping functions. The IRR of 1.07 for 'Linear vs. Concave-Down' indicates that trials governed by a linear shaping function had, on average, 1.07 as many clicks as those governed by a concave-down function. The results indicate that,

**Table 1. Descriptive statistics for AA and total clicks outcome measures.**

| | AA | | Total Clicks | |
|---|---|---|---|---|
| Comparison Group | Mean | Standard Deviation | Mean | Standard Deviation |
| All Functions | 31.44 | 4.57 | 25.40 | 10.07 |
| Linear | 31.45 | 4.04 | 25.54 | 9.60 |
| Concave Down | 29.17 | 4.18 | 27.28 | 10.33 |
| Concave Up | 33.69 | 4.39 | 23.38 | 9.90 |

**Table 2. Fixed effects from mixed effects models for AA and total clicks outcome measures.** The reported coefficient uses the first group as the referent level. For instance, for "Linear vs. Concave Down," linear is the referent level. IRR = Incidence Rate Ratio (IRR).

| Comparison Group | AA | | Total Clicks | |
|---|---|---|---|---|
| | Coefficient ($\beta$) | Confidence Interval | IRR ($e^{\beta}$) | Confidence Interval |
| Linear vs. Concave Down | -2.28 | (-3.24, -1.32) | 1.07 | (1.04, 1.10) |
| Linear vs. Concave Up | 2.25 | (1.29, 3.21) | 0.92 | (0.88, 0.95) |
| Concave Down vs. Concave Up | 4.52 | (3.56, 5.48) | 0.86 | (0.82, 0.89) |

relative to linear shaping functions, the concave-down shaping function produced a smaller AA and larger number of clicks. The relationship was reversed for the concave-up shaping function, which produced a higher AA and fewer clicks per trial. Compared to the concave-down shaping function, the concave-up function produced a larger AA and took resulted in a rate of clicks that decreased by a factor of 0.85.

The fitted variance components in the mixed effect regression models for the AA outcome allows conclusions to be drawn about the source of variance within observations and serves as a measure of effect size. A 0.63 proportion of the total variance was due to between subject differences, while 0.37 was due to the shaping functions.

## 4. Discussion

The results of the experiment summarized in this paper indicate that when shaping a target clicking behavior via an automatic process, the best results are achieved by using a concave-up shaping-function that rapidly narrows the window of reinforced behaviors towards a target outcome (illustrated in Fig 1b). A linear shaping function was the next most effective, followed by a concave-down shaping function, which was the least effective of the three alternatives that were considered. These empirical results are consistent with those from the previously developed computational model of automatic behavior shaping [12], yet there is some nuance within this corroboration. In most trials within the target click experiment, a full steady state, where 100% of clicks were within the target circle, was eventually reached. This was not the case in the computational model, where behaviors outside of the target class typically continued to be emitted throughout the entire experiment, although the probability of this occurrence varied with time. As a result, the best-performing computational shaping parameters changed as time progressed such that concave-up functions produced optimal results early in the experiment, while linear shaping functions performed best later. Therefore, the results of this validation experiment are most similar to shorter duration trials within the computational setting and suggest that it may be beneficial to reduce the duration of future computational experiments to best match actual behavior.

To enable future research into automatic shaping, the computational model was developed as a digital tool whose findings hopefully generalized to a laboratory experiment and eventually real-world trials, so that potential strategies for automatic behavior shaping could be evaluated before being deployed in the field. The results detailed herein partially validate this computational model, making it a potentially useful tool for intervention design. For example, in the Introduction we outlined a scenario where smokers could be rewarded for abstinent time periods, with the time required for reinforcement steadily increasing. And we posed the question of how quickly the reinforcement threshold should change. The finding from the computational model, that concave-up shaping functions best induce shaping, now verified by the experimental model in this study, suggests the threshold should rapidly increase at the outset of the intervention (e.g., 30-min per day over the first week) rather than more gradually. This

further suggests that shaping in other contexts (e.g. requiring progressively more daily physical activity to receive reinforcement) would also benefit from aggressively tightening reinforcement criteria at the beginning of the study, despite the risk that participants may struggle to perform the target behavior. Future trials should empirically investigate this question within a health intervention setting.

Several questions regarding the computational remain to be explored. First, the target behaviors in both the experimental and computational models involved a simple, unidimensional measure of distance–differences between screen locations in the former and integers for the latter. Many health behaviors (e.g., healthy eating, non-smoking) do not have a simple distance-measure component, and it is unclear how well the computational model will generalize to these behaviors. Second, when cultivating health behaviors, people rarely proceed in a linear fashion without experiencing missteps. For example, participants may skip a few weeks when trying to build a regular exercise routine or relapse when attempting to quit smoking. Ideally, a shaping routine should dynamically react to such occurrences; but neither the experimental shaping procedures outlined in this report, nor the computational model on which it is based, are equipped to do so. Lastly, since they commonly occur in naturalistic settings, the effects of variable reinforcement schedules that do not reinforce every instance of a target behavior remain to be explored. Such refinements must be built into future iterations of a computational model that aids with the design of automatic shaping systems. Despite these shortcomings, the findings from this study lend credence to the computational approach and encourages the development and testing of these modeling enhancements.

As technology continues to penetrate everyday living, health behavior interventions will be increasingly characterized by intensive real-time measurement along with dynamic feedback systems. Researchers have recognized that such interventions must be guided by dynamic, regulatory, and adaptive models of behavior change [20], a set of criteria that is satisfied by operant theory [11], which has a been proposed as a unifying approach to health-promotion science [21]. Therefore, as interventions increase the scope and accuracy with which behavior is observed, there will be increased opportunities to introduce operant science principles, such as automatic behavior shaping. While this is an unprecedented opportunity to increase the precision and reach of intervention science, there are several challenges associated with implementing automatic shaping routines, chief among these being the ability to objectively measure behavior and provide continual feedback. Such procedures require ample planning to navigate technological and operational hurdles, as well as forethought concerning the potential intrusiveness of ongoing participant monitoring. A related obstacle is the potential for participants to negatively react to long-term observation of their behavior, which might be viewed as surveillance. Consequently, it is critical that automatic shaping trials follow recommendations for e-health trials to include human support that is perceived as being trustworthy, benevolent, and having expertise [22]. Furthermore, we recognize that the automatic systems will not be able to replicate the nimble and complex processes that teachers, coaches, counselors, and other professionals regularly execute to shape behavior. However, if automatic shaping systems can capture a rudimentary approximation of human-implemented procedures and achieve some level of their success, the scalability offered by mHealth tools [23] has the potential to enable significant gains in public health. One can imagine the automatic monitoring and shaping of several health behaviors including exercise intensity and duration, bouts of non-smoking, caloric intake, and screen time, to name but a few.

Since behavior shaping is a relatively underutilized approach, it is interesting to compare it with nudging, another emerging psychological methodology that is becoming increasingly popular for promoting healthy behavior. Whereas shaping protocols manipulate the criteria under which reinforcement is provided, nudging consists of subtle, but precisely tuned, stimuli

designed to bias decisions in a beneficial direction [24]. For example, smaller plates at a buffet may be used to encourage reduced portions sizes [25]. While a nudge is typically composed of a single, discrete stimulus and strives to alter a behavior indirectly, shaping consists of more direct reinforcement scheme focused on approximations to a target. A potential moderator for the effect of both shaping and nudging is their relation to awareness. In the buffet example above, an individual will likely often fail to discriminate that plate sizes have changed, but there are other nudges where participants will directly perceive a stimulus (e.g., labelling a trash receptacle as landfill to reduce waste). In shaping, one would likely be aware that a behavior is being reinforced, but may or may not recognize a change in reinforcement criteria. Awareness of stimuli may influence the effectiveness of these behavior-change techniques and should be examined in future research. Additionally, it may be possible to shape nudges (e.g., successively smaller plates at the buffet), which should be explored.

There are several limitations to the study summarized in this report. First, the target behavior was relatively simple and had no direct bearing to tasks that one would typically encounter in daily life. Second, resource constraints allowed us to only investigate 3 shaping functions, while the computation model examined 574 distinct functions. We abandoned variation of the initial shaping window and instead chose representatives of three different functional shapes; it is possible that this coarse selection of shaping functions obscured subtle findings. We also did not include a no-shaping arm in the experiment, although the results of the computational model indicated that all shaping functions outperformed a no-shaping scenario. Additionally, the reinforcer used in this trial was a conditioned stimulus (pleasant sound/harsh sound) with no connection to real-world consequences, which might have impacted its ability to affect behavior. Lastly, the sample for this experiment consisted of undergraduate students at a single university, so the results may not generalize to the larger population. Future work should address these limitations.

In conclusion, the work within this paper describes a behavior-shaping experiment that was designed to mirror a previously performed computational study of shaping. The results indicate that the computational model's predictions about the supremacy of concave-up shaping functions are borne out in a laboratory setting. The extent to which these shaping results can be extended, both computationally and in the real-world, is a key question for future research. This point notwithstanding, this research helps lay the groundwork for scaling interventions by optimally using real-time sensing technology to shape health behavior.

## Acknowledgments

The authors thank the research participants for participating in this study.

## Author Contributions

**Conceptualization:** Vincent Berardi, Uri Maoz.

**Data curation:** Vincent Berardi.

**Formal analysis:** Vincent Berardi.

**Investigation:** Vincent Berardi, Uri Maoz.

**Methodology:** Vincent Berardi, Tian Lan, Uri Maoz.

**Project administration:** Vincent Berardi, Ariane Guirguis.

**Software:** Tian Lan, Uri Maoz.

**Supervision:** Vincent Berardi, Uri Maoz.

**Visualization:** Vincent Berardi.

**Writing – original draft:** Vincent Berardi.

**Writing – review & editing:** Tian Lan, Ariane Guirguis, Uri Maoz.

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
