## [Decision Letter · Decision Letter 0]

15 May 2024

PONE-D-23-36231Empirically Validating a Computational Model of Automatic Behavior ShapingPLOS ONE

Dear Dr. Berardi,

Thank you for submitting your manuscript to PLOS ONE. After careful consideration, we feel that it has merit but does not fully meet PLOS ONE’s publication criteria as it currently stands. Therefore, we invite you to submit a revised version of the manuscript that addresses the points raised during the review process.

We look forward to receiving your revised manuscript.

Kind regards,

Baeckkyoung Sung, Ph.D.

Academic Editor

PLOS ONE

3. Please remove your figures from within your manuscript file, leaving only the individual TIFF/EPS image files, uploaded separately. These will be automatically included in the reviewers’ PDF.

Additional Editor Comments:

The manuscript should be thoroughly revised in accordance to the reviewers' comments. Please clarify the novelty of the work and how it is related to mobile sensing technology. It is recommended for the authors to increase the data size (i.e., number of participants), if it is possible.

Reviewers' comments:

Reviewer's Responses to Questions

**Comments to the Author**

1. Is the manuscript technically sound, and do the data support the conclusions?

Reviewer #1: Yes

Reviewer #2: Yes

2. Has the statistical analysis been performed appropriately and rigorously? 

Reviewer #1: Yes

Reviewer #2: Yes

3. Have the authors made all data underlying the findings in their manuscript fully available?

Reviewer #1: Yes

Reviewer #2: Yes

4. Is the manuscript presented in an intelligible fashion and written in standard English?

Reviewer #1: Yes

Reviewer #2: Yes

5. Review Comments to the Author

Reviewer #1: The paper is well written and novel work to understand the concept of behavioral shaping

I have few minor comments:

1. In Figure 2 (a) Trial 1, 2, 3 and 45, it was accompanied with captions Linear, Concave Up, Concave Down, etc. However, I could not find any difference between the targets in different screen. Could you please clarify? Also, why 45 trials were chosen, is this enough? Also, was the targets were mixed with shaping function among the 45 trails, or it was continuous, like first 15 is linear, second 15 is concave up, etc.? Because the Figure 2(a) confused it up.

2. Also, I do not understand why would the participants wants to click 10 times to 50 times? As I do not feel the target is too difficult to select. Just a more clarification is needed on that.

3. Page 15, first line 1, add period after “the AA models)”

4. I was initially not sure still how this experiment can translate to real-world setting specifically if that happens with people with certain disorders. But the discussion section made justice to my doubts with good examples. Since these examples came at the end of the paper, it would be ideal to have a line with multiple example with (each with one word) in the abstract, so that the readers are aware how this experiment can be helpful and continue to read till discussion.

Reviewer #2: This study tests a computational model's predictions by designing a laboratory experiment to validate the effectiveness of automatic behavior shaping. The research particularly explores the impact of concave-up, linear, and concave-down shaping functions on participants' clicking behavior.

1. More detailed background information could be provided on how automatic behavior shaping specifically differs from traditional operant conditioning or other similar interventions. This will provide better context for readers less familiar with the field.

2. Highlight how this model and its validation can directly impact mHealth interventions, specifically elaborating on tangible applications.

3. Consider adding a justification for using auditory tones as reinforcers instead of tangible rewards or penalties, to address concerns that this reinforcement may not fully replicate real-world scenarios.

4. Provide further details on the interpretation of the incidence rate ratio for clicks and its practical implications for behavior shaping.

5. Consider expanding on the rationale behind the selection of 54 participants and the specific randomization strategy used. This helps clarify whether the sample size provides adequate power for detecting differences between shaping functions.

6. Elaborate on why three specific shaping functions (concave-up, linear, concave-down) were selected and how they represent a comprehensive spectrum of reinforcement criteria.

7. Clarify specific recommendations for practitioners or researchers looking to implement automatic behavior shaping in different settings, including potential limitations or challenges.

6. PLOS authors have the option to publish the peer review history of their article (what does this mean?). If published, this will include your full peer review and any attached files.

Reviewer #1: **Yes: **Pradeep Raj Krishnappa Babu

Reviewer #2: **Yes: **Junqiao Zhang

---

## [Author Response · Author response to Decision Letter 0]

24 Jul 2024

Responses to reviewer comments have been uploaded as a Cover Letter with the file name "Response to Reviewers.docx"

---

## [Decision Letter · Decision Letter 1]

9 Sep 2024

PONE-D-23-36231R1Empirically Validating a Computational Model of Automatic Behavior ShapingPLOS ONE

Dear Dr. Berardi,

Thank you for submitting your manuscript to PLOS ONE. After careful consideration, we feel that it has merit but does not fully meet PLOS ONE’s publication criteria as it currently stands. Therefore, we invite you to submit a revised version of the manuscript that addresses the points raised during the review process.

We look forward to receiving your revised manuscript.

Kind regards,

Baeckkyoung Sung, Ph.D.

Academic Editor

PLOS ONE

Additional Editor Comments:

The authors are suggested to thoroughly revise the manuscript in accordance to the comments by Reviewer 2.

Reviewers' comments:

Reviewer's Responses to Questions

**Comments to the Author**

1. If the authors have adequately addressed your comments raised in a previous round of review and you feel that this manuscript is now acceptable for publication, you may indicate that here to bypass the “Comments to the Author” section, enter your conflict of interest statement in the “Confidential to Editor” section, and submit your "Accept" recommendation.

Reviewer #1: All comments have been addressed

Reviewer #2: All comments have been addressed

2. Is the manuscript technically sound, and do the data support the conclusions?

Reviewer #1: Yes

Reviewer #2: Yes

3. Has the statistical analysis been performed appropriately and rigorously? 

Reviewer #1: Yes

Reviewer #2: Yes

4. Have the authors made all data underlying the findings in their manuscript fully available?

Reviewer #1: No

Reviewer #2: Yes

5. Is the manuscript presented in an intelligible fashion and written in standard English?

Reviewer #1: Yes

Reviewer #2: Yes

6. Review Comments to the Author

Reviewer #1: (No Response)

Reviewer #2: 1. Highlight the practical implications of the findings for health behavior interventions, emphasizing how the concave-up shaping functions could be utilized in real-world applications.

2. Expand on the background information to provide a more comprehensive understanding of the significance of automatic behavior shaping in health interventions. Include recent advancements and challenges in the field.

3. Clearly articulate the gap in current research that this study aims to fill. Specify why validating the computational model with human subjects is critical for advancing the application of behavior-shaping technologies.

4. State the research objectives explicitly, highlighting the importance of translating computational models into experimental settings.

5. Provide more detailed information about the participants, such as demographic characteristics and any relevant inclusion or exclusion criteria. This will help readers understand the sample's representativeness.

6. Elaborate on the experimental setup, including how the target and reinforcement circles were programmed and the rationale behind choosing specific shaping functions. Consider including a diagram or flowchart to illustrate the experimental procedure.

7. Provide a more detailed explanation of the statistical methods used, including any assumptions made and how they were tested. Consider including additional analyses, such as effect size calculations, to enhance the interpretation of the findings.

8. Discuss the results in the context of existing literature. Highlight how the findings support or challenge previous studies and the implications for designing automated behavior-shaping interventions.

7. PLOS authors have the option to publish the peer review history of their article (what does this mean?). If published, this will include your full peer review and any attached files.

Reviewer #1: **Yes: **Pradeep Raj Krishnappa Babu

Reviewer #2: No

---

## [Author Response · Author response to Decision Letter 1]

19 Oct 2024

Please see attached letter with responses to reviewer comments.

---

## [Decision Letter · Decision Letter 2]

4 Nov 2024

Empirically Validating a Computational Model of Automatic Behavior Shaping

PONE-D-23-36231R2

Dear Dr. Berardi,

We’re pleased to inform you that your manuscript has been judged scientifically suitable for publication and will be formally accepted for publication once it meets all outstanding technical requirements.

Kind regards,

Baeckkyoung Sung, Ph.D.

Academic Editor

PLOS ONE

Additional Editor Comments (optional):

The revised manuscript has well responded to the reviewer's comments.

Reviewers' comments:

Reviewer's Responses to Questions

**Comments to the Author**

1. If the authors have adequately addressed your comments raised in a previous round of review and you feel that this manuscript is now acceptable for publication, you may indicate that here to bypass the “Comments to the Author” section, enter your conflict of interest statement in the “Confidential to Editor” section, and submit your "Accept" recommendation.

Reviewer #2: All comments have been addressed

2. Is the manuscript technically sound, and do the data support the conclusions?

Reviewer #2: Yes

3. Has the statistical analysis been performed appropriately and rigorously? 

Reviewer #2: Yes

4. Have the authors made all data underlying the findings in their manuscript fully available?

Reviewer #2: Yes

5. Is the manuscript presented in an intelligible fashion and written in standard English?

Reviewer #2: Yes

6. Review Comments to the Author

Reviewer #2: The manuscript successfully meets publication criteria by clearly demonstrating the translation of a computational model of behavior shaping into an experimental human setting. Using mobile sensing technology, the study examines various shaping functions—concave-up, linear, and concave-down—finding that concave-up most effectively shapes target behavior. The experimental design, participant engagement, and outcome measures align with prior computational predictions, validating the model’s potential real-world applicability. The concise and structured presentation allows easy adaptation to health behavior interventions, including exercise, smoking, dietary habits, and screen time. The study’s implications make it a valuable contribution to behavior shaping research.

7. PLOS authors have the option to publish the peer review history of their article (what does this mean?). If published, this will include your full peer review and any attached files.

Reviewer #2: No

---

## [Editor Report · Acceptance letter]

19 Nov 2024

PONE-D-23-36231R2 

PLOS ONE

Dear Dr. Berardi, 

I'm pleased to inform you that your manuscript has been deemed suitable for publication in PLOS ONE. Congratulations! Your manuscript is now being handed over to our production team.

Kind regards, 

on behalf of

Dr. Baeckkyoung Sung 

Academic Editor

PLOS ONE